# Examining the Ugandan health system's readiness to deliver rheumatic heart disease-related services

Emma Ndagire[1,2], Yoshito Kawakatsu[3], Hadija Nalubwama[1], Jenifer Atala[1], Rachel Sarnacki[2], Jafesi Pulle[1], Rakeli Kyarimpa[1], Rachel Mwima[1], Rosemary Kansiime[1], Emmy Okello[1], Peter Lwabi[1], Andrea Beaton[4,5], Craig Sable[2,6], David Watkins[3,7] *

1 Uganda Heart Institute, Kampala, Uganda, 2 Children's National Hospital, Division of Cardiology, Washington, District of Columbia, United States of America, 3 Department of Global Health, University of Washington, Seattle, Washington, United States of America, 4 Cincinnati Children's Hospital Medical Center, Cincinnati, Ohio, United States of America, 5 Department of Pediatrics, University of Cincinnati College of Medicine, Cincinnati, Ohio, United States of America, 6 George Washington University School of Medicine, Washington, District of Columbia, United States of America, 7 Division of General Internal Medicine, University of Washington, Seattle, Washington, United States of America

* davidaw@uw.edu

**Data Availability Statement:** Survey data are available at https://doi.org/10.7910/DVN/SAKT55, and health worker interview transcripts are available at https://doi.org/10.7910/DVN/CYRR9O.

## Abstract

### Background

In 2018, the World Health Assembly mandated Member States to take action on rheumatic heart disease (RHD), which persists in countries with weak health systems. We conducted an assessment of the current state of RHD-related healthcare in Uganda.

### Methodology/Principal findings

This was a mixed-methods, deductive simultaneous design study conducted in four districts of Uganda. Using census sampling, we surveyed health facilities in each district using an RHD survey instrument that was modeled after the WHO SARA tool. We interviewed health workers with experience managing RHD, purposively sampling to ensure a range of qualification and geographic variation. Our final sample included 402 facilities and 36 health workers. We found major gaps in knowledge of clinical guidelines and availability of diagnostic tests. Antibiotics used in RHD prevention were widely available, but cardiovascular medications were scarce. Higher levels of service readiness were found among facilities in the western region (Mbarara district) and private facilities. Level III health centers were the most prepared for delivering secondary prevention. Health worker interviews revealed that limited awareness of RHD at the district level, lack of diagnostic tests and case management registries, and absence of clearly articulated RHD policies and budget prioritization were the main barriers to providing RHD-related healthcare.

### Conclusions/Significance

Uganda's readiness to implement the World Health Assembly RHD Resolution is low. The forthcoming national RHD strategy must focus on decentralizing RHD diagnosis and

**Funding:** This work was supported by the American Heart Association (grants 17SFRN33670611 [DW] and 17SFRN33630027 [CS]), https://professional.heart.org/professional/ResearchPrograms/StrategicallyFocusedResearchPrograms/UCM_454438_Strategically-Focused-Research-Networks.jsp. The funders had no role in study design, data collection and analysis, decision to publish, or preparation of the manuscript.

**Competing interests:** The authors have declared that no competing interests exist.

prevention to the district level, emphasizing specialized training of the primary healthcare workforce and strengthening supply chains of diagnostics and essential medicines.

## Author summary

Rheumatic heart disease is a major cause of cardiovascular death and disability, especially in less-developed parts of Africa and Asia. Unfortunately, medications and surgical procedures to prevent or treat rheumatic heart disease and its precursor rheumatic fever are greatly under-used, even though they are very effective. This study conducted an assessment of health system gaps in delivery of RHD related care in Uganda. We used quantitative and qualitative methods to find out, firstly, what percentage of health facilities are currently providing rheumatic heart disease services and, secondly, what things need to be fixed in order to improve service availability. We found that only 1–2% of health facilities are currently fully equipped to provide rheumatic heart disease prevention and treatment. The two biggest problems are that frontline health workers know very little about the condition and that the tests used to diagnose rheumatic fever and rheumatic heart disease are not widely available in many districts. Our assessment can serve as a baseline in monitoring the implementation of future interventions in Uganda. We are making our methods and tools publicly available so that ministries of health in other countries can use them to develop or expand their rheumatic heart disease programs.

## Introduction

Rheumatic heart disease (RHD) has nearly been eradicated in wealthy nations but remains a significant public health concern in low-resource countries like Uganda.[1] A recent study demonstrated no significant decline in mortality due to RHD in low-income countries over the last 25 years.[2] RHD can be prevented through evidence-based clinical interventions including (i) prompt treatment of group A streptococcal pharyngitis, an acute infection that incites the development of RHD ("primary prevention") and (ii) continuous antibiotic prophylaxis for individuals with a history of rheumatic fever or early-stage RHD ("secondary prevention"). RHD and its precursor condition rheumatic fever are also strongly linked to determinants of communicable diseases (e.g., poor sanitation and overcrowding within homes and schools), the remediation of which is sometimes called "primordial prevention." Finally, advanced medical and surgical interventions are also options for individuals who acquire RHD, though access remains very low in many countries.[3]

The impact of community-based primary and secondary prevention measures in low- and middle-income countries has been well-established: the burden of RHD can be reduced by more than 90% within a decade of intervention scale-up.[4,5] WHO guidance over the past several decades has emphasized that primary and secondary prevention efforts are the cornerstone of the public health response to RHD. Experts generally agree that RHD persists in part because of weak primary healthcare systems that are failing to deliver evidence-based care to those in need, most of whom come from marginalized populations like rural communities and urban informal settlements.[6] Low availability of RHD preventive interventions and treatments can be remediated by health system reforms including the decentralization of preventive care, formation of national control programs, and focused expansion of tertiary services.[7] These measures have not occurred in most low-resource countries: until recently, RHD has not been part of national health agendas or strategies, despite its intersection with priorities such as child and adolescent health, maternal health, and non-communicable diseases. The

71[st] World Health Assembly (WHA) adopted a Resolution on RHD in 2018, mandating Member States to prioritize RHD control programs.[8] In order to implement the Resolution, policy makers need up-to-date information on the status of RHD prevention and control measures in their countries. In the African context, most of the local evidence relating to RHD has focused on clinical and epidemiological aspects of the disease but has not systematically identified or quantified local health system gaps or priorities for intervention design.

In recent years, the Uganda Heart Institute (UHI), a semi-autonomous, publicly-funded clinical and research facility in Kampala, has engaged with the Uganda Ministry of Health to develop a national RHD strategy in response to the WHA Resolution. The current study describes a situational assessment that we conducted to understand the current state of RHD-related health services in Uganda. As this study was the first of its kind related to RHD, we aimed to develop an approach and tools that could not only serve as an input to the policy formulation process in Uganda but also be applied to other countries seeking guidance on implementation of the WHA Resolution.

## Methods

### Ethics statement

Ethics approval was obtained from Makerere University School of Medicine Research and Ethics Committee (REC RF 2018–082), University of Washington Human Subjects Division (STUDY00002855), and from the Uganda National Council for Science and Technology (SS 5081). Permission was sought from relevant authorities within the districts prior to conduct of health facility surveys. Written informed consent was obtained from all health workers that were interviewed.

We designed this study as a mixed methods deductive simultaneous design with a core quantitative component and a supplemental qualitative component. The primary rationale for using mixed methods was complementarity: we anticipated that the quantitative component would demonstrate significant gaps in RHD related care, so we added the qualitative component to elaborate on and illustrate the quantitative findings. All data were collected between July 2018 and December 2019.

### Study sites

At the time of this study, Uganda had a population of over 40 million, with over 80% of the population residing in rural areas and 38% earning less than $US 1 per day.[9] We sought to do a comprehensive assessment of RHD-related healthcare in four districts, one in each of the four main administrative regions (Fig A in S1 Text). Districts were chosen (i) because they were generally representative of the epidemiological, demographic, and economic situations in each of their regions, and (ii) because rheumatic fever epidemiology studies were being done concurrently in the same districts. Lira district (population 410,000) is home to one of the largest cities in the Northern region. Wakiso district (population 2.0 million) is adjacent to Kampala in the Central region and is the second most populated district in the country. The other two districts we selected were Mbarara district (population 470,000) in the Western region and Tororo district (population 520,000) in the Eastern region. All four districts had a mix of urban and rural administrative units.

### Health system organization in Uganda

Uganda's health system is structured into public and private sectors. The private sector includes private for-profit and private not-for-profit (i.e., faith/charity-based) practitioners.

Health services are decentralized within districts and are organized hierarchically (Table A in S1 Text). Publicly-owned health facilities are the main providers of primary health care services, and the primary healthcare workforce is comprised mostly of nurses and midwives. The Uganda Heart Institute, located in Kampala, is the sole public-sector provider of specialized cardiovascular services.

## Survey procedures

Since we sought to conduct a comprehensive assessment of health facility readiness within each district, we used the census method of sampling. A total of 501 health facilities were included on the official district health office lists. We excluded 93 privately-owned facilities that were included on these lists but were non-functional (12 in Lira, 43 in Wakiso, three in Mbarara, and 35 in Tororo). In addition, three health facilities in Wakiso district were exceptionally difficult to access due to poor roads. The remaining 405 public and private health facilities on the district health office facility censuses were invited to participate in the survey. Each facility was asked to send a representative to an orientation meeting at which the project research staff gave a formal presentation of the study objectives and design and answered questions. Facility representatives who agreed to participate were assigned a survey day on which a research nurse collected data face-to-face. One private facility in Tororo district declined to participate, and two other facilities (health centers II) in Tororo district were excluded during the data analysis phase because of incomplete data. The final sample we used for analysis was comprised of 402 out of the 408 functional health facilities in these four districts (i.e., 98.5% survey coverage).

The WHO Service Availability and Readiness Assessment (SARA) tool[10] was used to assess general service availability in each health facility. We developed an RHD facility assessment tool (S1 Text), modeled after the categories and styles of questions used in SARA. RHD-specific questions were designed as sub-modules that were integrated within each of the SARA modules. Questions assessed the availability of key inputs to primary prevention, secondary prevention, and RHD medical management. The three broad input categories were guidelines/training, medications, and diagnostic tests. The list of specific indicators we constructed to measure RHD service availability and readiness is provided in Table B in S1 Text.

Data collection included verbal responses as well as direct observation at the health facility. Research nurses read the survey questions to respondents and confirmed responses by performing a visual assessment of availability of the services, medicines, and diagnostics that the survey participant indicated were available at the site. Respondents for each type of question included facility "in-charge" nurses or clinical officers, pharmacy staff, or nurses as appropriate, with the most knowledgeable person providing responses to relevant questions. (For example, pharmacy staff provided responses to questions regarding medication availability.) Participation was voluntary, and there was no penalty for declining to participate. Research nurses recorded responses onto a mobile REDCap database maintained by Children's National Hospital in Washington, DC.[11] Data were checked for correctness and errors on a daily basis by two members of the research team (EN, JA).

## Quantitative data analysis

Survey responses were first analyzed using descriptive statistics. We looked at proportions of facilities that responded in the affirmative to each item on the survey, both at the district level and stratified by facility characteristics. We then constructed a composite score of service readiness for primary prevention, secondary prevention, and RHD medical management separately and in combination. Each of these services was mapped to the facility types and level of

the health system at which it could be reasonably delivered. For example, all facilities would be expected to be able to deliver primary prevention, whereas only health centers IV and hospitals would be expected to deliver medical management of RHD complications. The "total" score for each facility thus reflected the service(s) that it would be able to provide based on its level within the health system (see Table B in S1 Text). These scores were compared against general health facility service readiness scores from the SARA surveys, which we computed using procedures specified in the SARA handbook [12].

We then explored which facility characteristics were associated with higher RHD readiness scores. We used negative binomial regression models that included the following covariates: managing authority, level of health facility, district, number of nurses, and location of health facility (rural/urban). (Negative binomial models, rather than Poisson models, were used due to overdispersion.) An offset was used for the dependent variable, total RHD readiness score. The highest RHD readiness scores that could be attained for health centers II, III, IV and hospitals were 5, 6, 8, and 11, respectively. All quantitative data analysis was performed in R version 4.0 (R Foundation for Statistical Computing, Vienna, Austria, 2020; https://www.R-project.org).

### Health worker interview procedures

To provide additional insight into the quantitative results, we conducted interviews with 36 health workers who were likely to interact with patients with RHD. Prior experience of the research team working in these districts suggested that most primary healthcare staff were unaware of rheumatic fever and RHD and therefore would not yield any informative data about health system barriers. A list of health workers at the highest-level health center or hospital in each district was obtained, and potential subjects were approached by the research staff and invited to participate. We purposively sampled physicians, midlevel care providers (clinical officers), nurses, and pharmacists to reflect a range of specialization. Interviewers had no connection to participants and therefore no obvious conflicts of interest. In contrast to the other three districts, the Tororo district had no previously diagnosed cases of RHD on its hospital registers. Since we did not expect health workers in this district to have any experience providing RHD care, we did not include health workers from this district in our qualitative sub-study.

Subjects were interviewed in English by a research nurse and a public health officer trained in qualitative interview methods. Translation to the participants' primary language was not necessary since English is an official language used in various institutions in Uganda. Each interview was conducted using a semi-structured guide that contained questions and prompts regarding potential enablers and barriers to delivery of RHD-related services (see S1 Text). Interviews were conducted at a private location/room at each health facility and usually lasted 30–60 minutes. All interviews were audio-recorded and transcribed verbatim.

### Qualitative data analysis

We took field notes during and immediately after the interviews to capture initial impressions and themes that were incorporated into the overall coding and analysis. Using a thematic analysis approach, we did both deductive and inductive coding to develop categories and themes. Two team members (HN, JA) independently coded the data and matched themes against a pre-written deductive codebook whose structure was informed by the socioecological model. Preliminary findings from the facility survey analysis also informed the coding process and aggregation of codes. The codebook was updated iteratively according to grounded theory and axial coding methods. Representative quotations were extracted from transcripts to illustrate

major codes and themes. A sample of the transcripts and codes were then reviewed by two independent team members (EN, RS) to confirm that codes had been classified correctly and that theoretical saturation had been achieved. Different dimensions and commonalities of each theme and their distribution across participant type/variables (e.g., health worker qualification) were also analyzed. In March 2020, most human subjects research in Uganda was suspended in the context of the novel coronavirus (SARS-CoV-2) pandemic, so we were not able to do participant checking as part of the data analysis. All qualitative data were managed and analyzed using Atlas.ti version 8.2.

## Results

As noted previously, we analyzed data from 402 health facilities, of which 50% were public. Wakiso district had the highest number of health facilities (38% of total sample). Fifty percent of facilities were located in rural areas (Table 1).

Proportions of facilities responding affirmatively to each RHD service readiness indicator are provided in detail in Table 2. Below, we summarize findings across the three types of RHD services and four districts and look at facility characteristics that predict overall RHD readiness.

### Primary prevention

Most respondents reported that their facilities had not undergone any training on management of pharyngitis in the past two years. The most extreme examples were health centers II-III, where only 1.3–1.5% of respondents reported receiving training. Yet when asked, over 70% of health centers III and higher were able to present our team with a copy of their clinical guidelines, which include guidelines for management of pharyngitis. Amoxicillin and benzathine penicillin G were widely available across all health facility levels. When we compared service readiness across districts, Wakiso district facilities had the highest mean readiness score for primary prevention overall, while Tororo district had the lowest (Fig B in S1 Text).

**Table 1. Characteristics of health facilities in the four districts included in this study.**

| Characteristics | HC II | HC III | HC IV | Hospital |
|---|---|---|---|---|
| **No. of health facilities** | 202 | 150 | 36 | 14 |
| **Managing authority, N (%)** | | | | |
| Government | 109 (54) | 67 (45) | 17 (47) | 6 (43) |
| NGO/Not-for-profit | 11 (5.4) | 13 (8.7) | 2 (5.6) | 1 (7.1) |
| Private-for profit | 80 (40) | 57 (38) | 16 (44) | 5 (36) |
| Mission/Faith-based | 2 (1.0) | 13 (8.7) | 1 (2.8) | 2 (14) |
| **No. of medical doctors, mean (SD)** | NA | NA | 3.4 (4.0) | 12 (17) |
| **No. of clinical officers, mean (SD)** | NA | 1.4 (0.96) | 2.8 (3.5) | 6.8 (8.3) |
| **No. of nurses, mean (SD)** | 2.3 (1.2) | 3.4 (2.0) | 9.0 (7.1) | 27 (30) |
| **District, N (%)** | | | | |
| Lira | 56 (28) | 30 (20) | 4 (11) | 1 (7.1) |
| Wakiso | 52 (26) | 77 (51) | 17 (47) | 8 (57) |
| Mbarara | 46 (23) | 17 (11) | 10 (28) | 1 (7.1) |
| Tororo | 48 (24) | 26 (17) | 5 (14) | 4 (29) |
| **Located in rural area, N (%)** | 108 (54) | 80 (53) | 13 (36) | 2 (14) |

Notes: HC- health center, NA- not applicable. "Hospital" includes both district hospitals and regional referral hospitals. (Mbarara and Lira districts included regional referral hospitals, whereas Wakiso and Tororo districts did not.) "Rural" refers to health facilities located outside urban centers.

**Table 2. RHD service readiness by facility type.**

|  | HC II | HC III | HC IV | Hospitals |
|---|---|---|---|---|
|  | 202 | 150 | 36 | 14 |
| Primary prevention indicators | N (%) | N (%) | N (%) | N (%) |
| National guidelines for pharyngitis management | 99 (49) | 105 (70) | 27 (75) | 11 (79) |
| Training of staff on pharyngitis management | 3 (1.5) | 2 (1.3) | 4 (11) | 2 (14) |
| Observed stock of oral amoxicillin | 176 (88) | 136 (91) | 33 (92) | 14 (100) |
| Observed stock of injectable benzathine penicillin | 73 (36) | 132 (88) | 33 (92) | 13 (93) |
| Observed stock of oral azithromycin | 65 (32) | 83 (55) | 27 (75) | 12 (88) |
| Secondary prevention indicators | N (%) | N (%) | N (%) | N (%) |
| National guidelines for RHD secondary prevention | NA | 12 (8.0) | 4 (11) | 3 (21) |
| Observed stock of benzathine penicillin* | NA | 132 (88) | 33 (92) | 13 (93) |
| Observed stock of azithromycin* | NA | 83 (55) | 27 (75) | 12 (86) |
| Availability of functional ultrasound machine | NA | NA | 10 (28) | 9 (63) |
| Availability of functional ECG machine | NA | NA | 4 (11) | 1 (7.1) |
| Availability of CRP test | NA | NA | NA | 0 (0.0) |
| Availability of ESR test | NA | NA | NA | 8 (57) |
| Availability of ASO Titer test | NA | NA | NA | 0 (0.0) |
| RHD medical management indicators | N (%) | N (%) | N (%) | N (%) |
| National guidelines for RHD management | NA | NA | 12 (33) | 2 (14) |
| Training of staff on RHD management | NA | NA | 7 (19) | 3 (21) |
| Training in RHD diagnosis (including ultrasound) | NA | NA | 5 (14) | 2 (14) |
| Observed stock of oral furosemide | NA | NA | 28 (78) | 12 (86) |
| Observed stock of parenteral furosemide | NA | NA | 29 (81) | 10 (71) |
| Observed stock of oral spironolactone | NA | NA | 8 (22) | 7 (50) |
| Observed stock of oral digoxin | NA | NA | 15 (42) | 10 (71) |
| Observed stock of any oral ACE inhibitor | NA | NA | 17 (47) | 11 (79) |
| Observed stock of any oral beta blocker | NA | NA | 11 (31) | 4 (29) |
| Observed stock of warfarin | NA | NA | NA | 2 (14) |
| Availability of functional ultrasound machine* | NA | NA | 10 (28) | 9 (63) |
| Availability of functional ECG machine* | NA | NA | 4 (11) | 1 (7.1) |
| Availability of INR test** | NA | NA | NA | 2 (14) |

Notes: ACE- angiotensin-converting enzyme, ASO titers- anti-streptolysin O titers. CRP- C-reactive protein, ECG- electrocardiography, ESR- erythrocyte sedimentation rate, INR- international normalized ratio, HC- health center, NA- not applicable, i.e., not expected to exist at a facility at this level of the system.

* Items that are included in more than one type of service; these are included in individual service readiness scores but are not double counted in the total score calculation.

** Refers to standard, laboratory-based INR testing; the survey also asked about point-of-care INR testing, which was not available at any hospital. Private facilities had more requisites (drugs and diagnostic tests) and in particular most of the facilities had at least one laboratory test for acute rheumatic fever.

## Secondary prevention

Guidelines for secondary prevention of RHD were only present at 21% of hospitals, 11% of health centers IV and 8% of health centers III. Diagnostic tests for rheumatic fever, which is a key component of secondary prevention, were scarce. Only 63% of hospitals and 28% of health centers IV had a functional ultrasound machine, and only one hospital (7%) had an electrocardiography machine. Almost no facilities were able to perform the blood tests that are used to diagnose rheumatic fever; the exception was the erythrocyte sedimentation rate test, which was available at some health centers IV and hospitals (predominately private). Fig 1 provides the mean readiness score for each district (around 50% except for Tororo, which was lower)

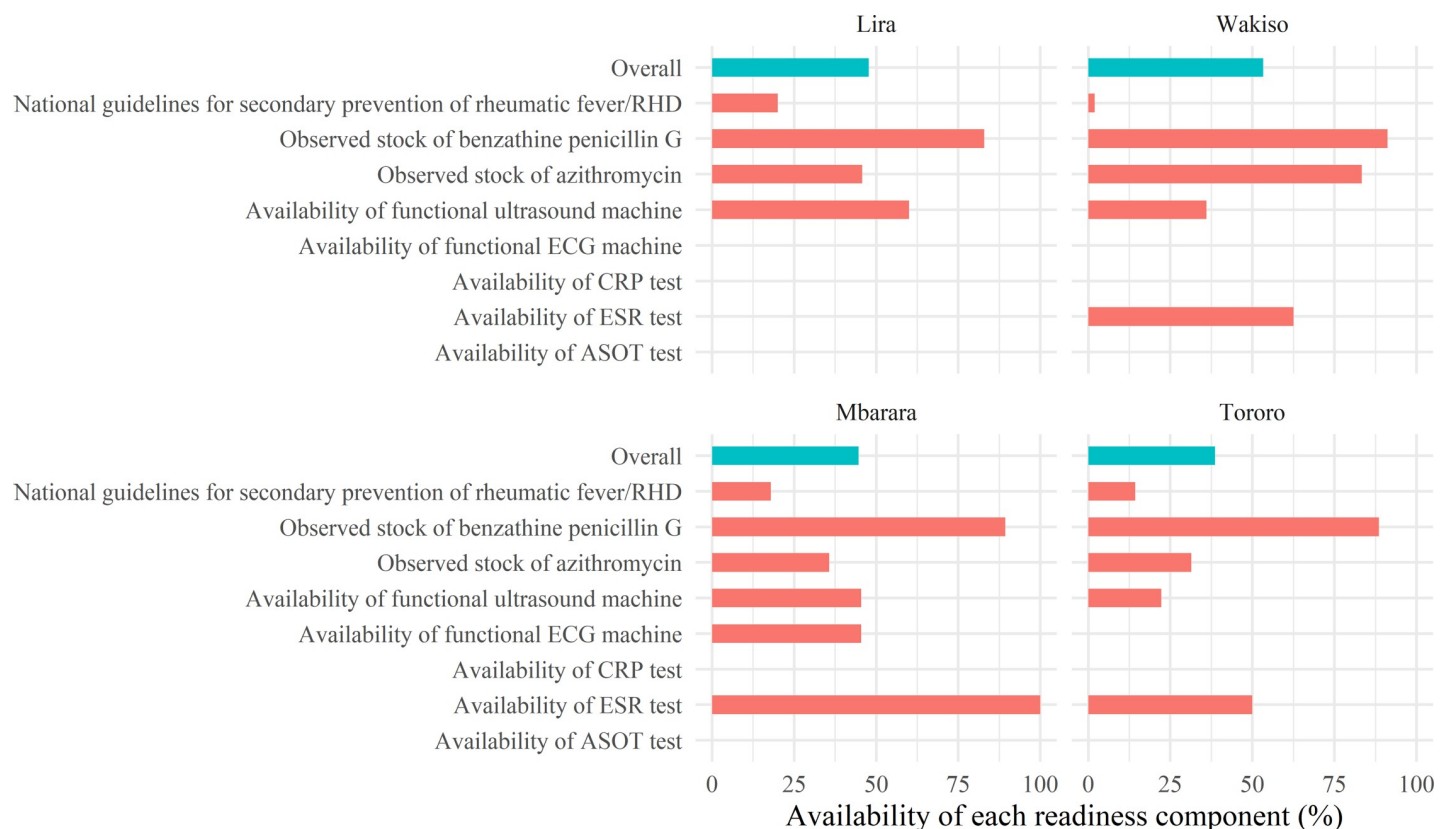

**Fig 1. Secondary prevention service readiness by district.** The figure provides estimates by district of the average percent availability of each indicator used to measure secondary prevention readiness. The overall percent availability of secondary prevention, i.e., the presence of all indicators at a given facility, is shown as a blue bar.

and illustrates the major secondary prevention "bottlenecks" (i.e., indicators with very low availability, significantly affecting the mean readiness score) in each district.

## RHD medical management

Less than 25% of health workers had received training on RHD in the past two years. Availability of medications for heart failure management ranged widely: >70% of health centers IV and hospitals had furosemide in stock, while <50% of health centers IV had digoxin or any angiotensin-converting enzyme inhibitor. Beta blockers were the least available of all cardiovascular medicines. Warfarin and standard, laboratory-based international normalized ratio (INR) testing were only available at 14% of hospitals, and point-of-care INR testing was not available at any hospital. Mbarara district had the highest mean readiness score (50%) while Tororo district had the lowest (25%) (Fig C in S1 Text).

## Overall RHD service readiness

Private facilities had higher RHD service readiness scores compared to public facilities across all districts. RHD medical management readiness scores were lower than those for primary and secondary prevention service readiness in all districts. Overall, Tororo district had the lowest total scores for RHD service readiness (Table C in S1 Text). Across the entire sample, the proportions of facilities that had all the requisite inputs for primary prevention, secondary prevention, and RHD medical management were 1%, 2%, and 2%, respectively (Table D in S1 Text).

When we compared RHD-specific readiness scores to general facility readiness scores calculated from the WHO SARA survey data from the same facilities, we found that facilities that had higher general readiness scores also had better RHD-specific service readiness scores (Fig D in S1 Text).

Finally, in a series of regression analyses, we looked at facility- and district-level predictors of overall RHD service availability and readiness. Table 3 displays the results of the final multivariate model. Compared to Lira district facilities, Wakiso and Mbarara district facilities had significantly higher readiness scores, whereas Tororo district facilities had non-significantly lower scores. Non-public facilities all had significantly higher readiness scores than government health facilities. Health centers III had higher RHD service readiness than health centers II (adjusted rate ratio 1.17, 95% CI: 1.02–1.35, *p*-value: 0.026), including inputs for both primary and secondary prevention. A higher number of nurses in a health facility was also associated with a significantly higher RHD readiness score.

## Health worker interviews: barriers to RHD-related care

Respondents identified a number of barriers and enablers to RHD care and provided a number of suggestions for improvement (Fig E in S1 Text). Three major themes emerged from the qualitative interviews, all of which provided additional insight into the survey findings: (i) knowledge gaps, (ii) logistical hurdles, and (iii) lack of articulated policies concerning RHD (Fig F in S1 Text). Limited knowledge of RHD was frequently mentioned by health workers and was attributed to lack of pre-service and in-service training. Health workers felt that knowledge gaps were leading to delayed identification of cases and management practices that deviated from accepted guidelines. As one participant stated,

**Table 3. Factors associated with RHD health service readiness in Uganda.**

| | Adjusted rate ratio | 95% Confidence Interval | *p* value |
|---|---|---|---|
| **District** | | | |
| Lira | Ref. | - | - |
| Wakiso | 1.23 | 1.04, 1.45 | 0.021 |
| Mbarara | 1.39 | 1.14, 1.71 | 0.001 |
| Tororo | 0.803 | 0.642, 1.00 | 0.065 |
| **Location** | | | |
| Urban | Ref. | - | - |
| Rural | 0.947 | 0.820, 1.09 | 0.49 |
| **Managing authority** | | | |
| Government | Ref. | - | - |
| NGO/not-for-profit | 1.27 | 0.993, 1.62 | 0.044 |
| Private-for-profit | 1.57 | 1.34, 1.85 | <0.001 |
| Mission/faith-based | 1.59 | 1.21, 2.07 | 0.001 |
| **Level of health facility** | | | |
| Health center II | Ref. | - | - |
| Health center III | 1.17 | 1.02, 1.35 | 0.026 |
| Health center IV | 0.983 | 0.807, 1.22 | 0.76 |
| Hospital | 0.880 | 0.630, 1.21 | 0.18 |
| **Number of nurses in health facility** | 1.01 | 1.00, 1.02 | 0.015 |

Notes: coefficients are derived from a negative binomial model including the covariates listed above, an intercept (not shown) and an offset dependent variable, total facility readiness score.

"*If the patient comes here with [RHD] and I personally don't know the signs and symptoms, I just continue giving antibiotics. . . it takes [a] long [time] to be diagnosed. . . [with] heart disease.*" (HW 005 Wakiso)

Health workers expressed a desire to be better trained in order to provide RHD care:

"*I feel I should be trained about [rheumatic fever and RHD]. . . so that I can take care of [these patients]. . . I [am] confident [that I can] take care of them, but I need the knowledge*" (HW 009 Lira)

Consistent with the facility survey data, one of the most frequent logistical hurdles that health workers reported was diagnostic testing for rheumatic fever and RHD. Lack of diagnostic testing has led to a situation where very few Ugandans with RHD have been identified by the health system and linked to care, despite a population prevalence of at least 1%.[13] Without diagnostic tests, health workers are not able to confirm diagnoses or assess for disease complications. In one participant's words,

". . . for the time I have been [working] here, I have not diagnosed any patient with rheumatic heart disease. It's not that I have not seen [them], I just can't confirm [the diagnosis]." (HW 012 Wakiso)

Participants also reported a lack of standardized RHD case management systems, impacting the care received by the few individuals who have been diagnosed and limiting the ability of districts to track new cases accurately. In the absence of clinical case registries (an approach endorsed by RHD experts internationally), the magnitude of the RHD problem is being underestimated in their districts, and hence administrators do not prioritize RHD in their budgets:

"Most [of the time] patients come, [and] we do all that we have to diagnose, investigate, [and] treat, [and then] they go home, and it is not captured anywhere. It creates an illusion [that] the patients are few. . . partly because rheumatic heart disease is not given special attention like HIV." (HW 010 Lira)

Participants also expressed concern over lack of national policies regarding RHD. They felt that, because of insufficient policy prioritization, funding for RHD services in their districts has remained poor:

"..right now. . . [the government] are not even aware that [RHD is] there. . . I have always sat on meetings, [and] we have always done the budget together, but there has never been any funding for RHD." (HW 008 Lira)

Fig 2 summarizes the factors (organized within the socioecological framework) that emerged from the qualitative data as critical barriers to delivery of RHD-related healthcare.

## Discussion

While RHD remains a significant public health problem in many low- and middle-income countries, historical experience has shown that the condition can be eliminated.[4,5] However, elimination requires a coordinated response and investment in the capabilities of primary healthcare workers to deploy primary and secondary prevention interventions at scale.[14] To our knowledge, this is the first in-depth examination of current healthcare delivery patterns

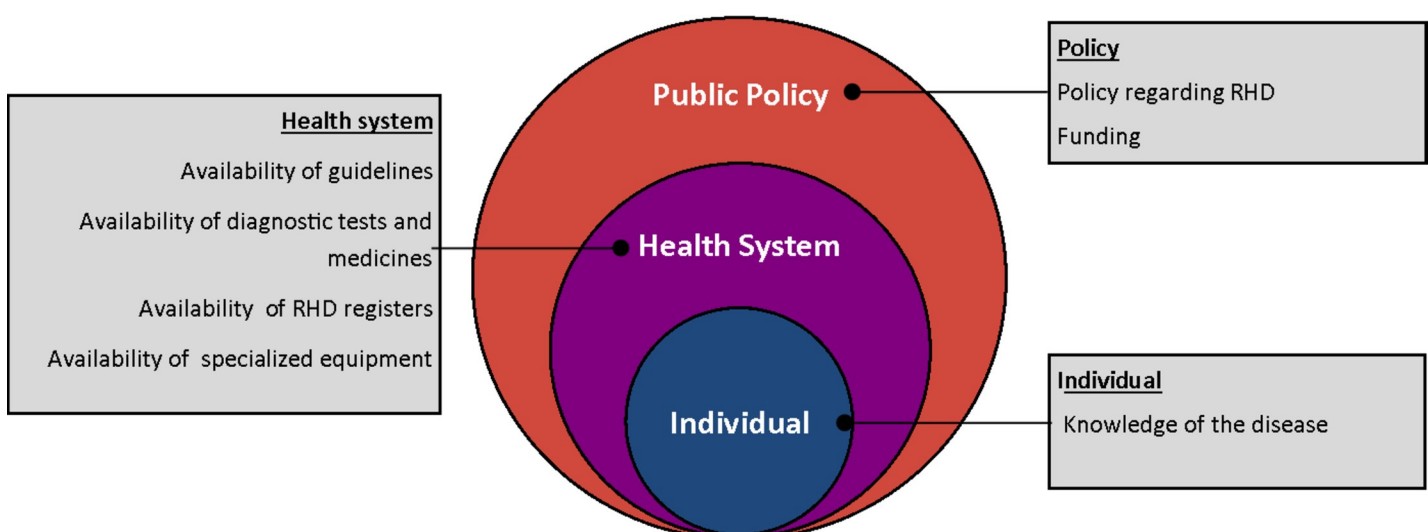

**Fig 2. Health worker perceptions of barriers to RHD care in Uganda.** The figure presents, within the socioecological model, the factors that health workers identified as key barriers to providing RHD prevention and treatment services.

and gaps related to RHD in a low-resource, RHD-endemic country. Our methods and data collection instruments (see S1 Text) could be used by researchers and practitioners in other countries who wish to conduct situational assessments and develop or refine their strategies for implementing the WHA Resolution on RHD.

We documented low availability of RHD-related services in four districts in Uganda, with intervention "coverage" being only 1% (for primary prevention) to 2% (for secondary prevention and RHD medical management) on average across the entire sample (Table D in S1 Text). By triangulating facility survey data and health worker interviews, we conclude that the most notable and urgent bottlenecks to remediate when planning Uganda's RHD programs are low provider training/awareness and lack of diagnostic tests.

To respond to an episode of pharyngitis, rheumatic fever, or RHD, frontline health workers must have the knowledge and skills required to diagnose and treat these interlinked conditions. Yet our study found that most health workers in Uganda remain unaware of these conditions. Our findings are similar to studies conducted in Sudan[15] and Zambia.[16] We note that, though the overall level of health worker awareness was low in our sample, the proportion of health workers that had received training was higher in Lira and Mbarara districts compared to Wakiso and Tororo districts. Awareness may have been increased due to concurrent RHD research activities in those two districts.[17] However, because the research programs were being conducted as epidemiological investigations without an explicit partnership with the ministry of health to scale up access to RHD-related care, we are confident that any influence the research had would be limited to provider awareness and would not extend to other components of care, such as availability of medicines and diagnostics at health facilities. While it is clear that training is needed, it is not known whether pre-service education, in-service education, or a combination of the two would lead to the most durable improvements in health worker knowledge and practice. Efforts to educate health providers and identify cases earlier, at the district level, could address one of the major determinants of poor outcomes: most patients experience delays in referral and only receive a definitive diagnosis at a tertiary hospital, often when their disease is too advanced for surgical intervention.[18]

To meet population demand for primary and secondary prevention of RHD, health facilities require antibiotics that are active against group A streptococcus, including narrow

spectrum penicillins or alternatives for individuals who are allergic to penicillin. In this study, we measured availability of benzathine penicillin G, amoxicillin, and azithromycin, which are all listed on WHO's Model Essential Medicines List and could be considered first-line for RHD prevention in African countries. It is not surprising that these drugs were widely available, since they have other clinical indications related to maternal and child health, which are priorities of governments and donors in low-income countries. Nonetheless, these drugs were less likely to be available at primary healthcare facilities, including at health centers III and especially at health centers II. On the other hand, penicillins are among the most prescribed antibiotics at primary health facilities in Uganda, so it is possible that there were stock-outs at a number of health centers at the time we conducted the study, reflecting natural variation in demand rather than persistent deficiencies in supply chains. More importantly, though, we did not assess the number of antibiotic doses available at each facility, so we cannot say whether the current supply chain would be robust enough to meet increased demand resulting from efforts to scale up access to RHD preventive services. In addition, only half of Ugandans with RHD maintain optimal adherence,[19] so efforts to improve adherence will also increase demand for medications and the likelihood of stockouts—if supplies remain at their current levels. An assessment of the current state of penicillin supply chains in Uganda, and projections of demand resulting from scale-up of RHD services, will be an important direction for future research.

Delivery of secondary prevention relies on identification of individuals who have a history of rheumatic fever or RHD and therefore requires capacity to diagnose both of these conditions at a district level. Standard criteria for diagnosis of rheumatic fever incorporate antistreptococcal titers or antigen tests, echocardiography, electrocardiography and acute phase reactant tests (Table 2). As demonstrated in this study, these diagnostics are nearly universally unavailable in these four districts of Uganda. Individuals who are counseled to get these tests in Uganda have to either travel to tertiary facilities[20] or in some cases pay out-of-pocket to get tests at private facilities, both of which are important barriers to delivering secondary prevention in socioeconomically disadvantaged and far-flung communities where RHD is more common. Our findings suggest that, in the short run, approaches to scaling up secondary prevention that rely entirely on "passive" identification of rheumatic fever cases are unworkable in Uganda. "Active" strategies that decentralize diagnosis (e.g., using handheld cardiac ultrasound in community or primary healthcare settings) could prove feasible and cost-effective,[21,22] though they do not obviate the long-term need for improvements in laboratory services. At the same time, alternative approaches to diagnosis of rheumatic fever are urgently needed for countries where community laboratory capacity is limited and is likely to remain so.

In Uganda, most individuals with RHD either die before reaching medical care or present to tertiary facilities with advanced disease requiring surgical interventions, and access to surgical services remains far lower than demand.[23] To improve outcomes of individuals with mild-to-moderate RHD and those on surgical waiting lists, district health facilities need to maintain a constant supply of cardiovascular medicines to manage RHD complications. While our study documented low availability of medicines for RHD-related heart failure, these same medicines are used for a variety of cardiovascular conditions, including hypertension, ischemic heart disease, atrial fibrillation, and stroke. Greater investment in supply chains of medicines for RHD could have a positive spillover effect on individuals receiving care for other cardiovascular diseases. However, to maximize value for money in pharmacological treatment for complications of RHD (like heart failure) health planners should simultaneously work to improve access to cardiac surgery, the definitive treatment for RHD.[24]

We found that the Tororo district in the Eastern region generally had the lowest scores on all indicators in this study. Compared to Lira and Mbarara, at the time of this study the Tororo

district had not served as a site for any RHD-related research projects or clinical activities (such as case registers) to date. In this way, Tororo is probably more like many of the other districts in Uganda that have limited or no RHD services, and therefore it is most likely to be representative of the national situation outside of Kampala, Lira, or Mbarara as well as other conditions in other countries in sub-Saharan Africa. The Wakiso district had an overall RHD service readiness score similar to that of Mbarara despite lack of dedicated RHD services. This may be explained by the district's proximity to the capital, a larger population and a larger number of private health facilities that have better RHD service readiness. Finally, we note that health centers III had better RHD service readiness scores as compared to health centers II, which suggests that health centers III are appropriate short- to medium-term targets for piloting and rolling out primary and secondary RHD prevention programs.

This study had several important limitations in addition to those discussed above. Most notably, our project was limited to four districts, sampled for strategic reasons rather than through the use of probability-based methods. A nationally representative survey of health facilities that employed formal survey sampling techniques would be able to provide estimates that would be "generalizable" to the entire country, whereas our study can only claim to reflect the experience in four districts. Further, the views of the health workers interviewed in this study may be different from those who are not included in the study. (We did not interview frontline primary healthcare workers, because we expected them to have little to no knowledge of or experience with rheumatic fever or RHD.) Finally, since this was a cross sectional study, our results represent a snapshot in time. Our results on medication availability should be interpreted with caution, since commodities supply chains are dynamic, and depending on the time of the month or year that a facility is surveyed, its medicines stock may vary considerably.

Our study should also be placed in the broader context of RHD-related health services research in Uganda. We focused on measuring the availability of inputs and resources needed for RHD-related healthcare. Other gaps in healthcare access and quality, which we did not attempt to ascertain in this study, would present additional challenges to achieving universal health care coverage in the context of RHD. Remediation of access barriers will not only require "supply-side" solutions (e.g., improving availability of healthcare inputs), which we emphasize here, but also "demand-side" solutions that address deterrents to health-seeking behavior. For example, ongoing research by our group has begun to shed light on the significant transportation barriers and catastrophic out-of-pocket costs faced by Ugandans with RHD.[25] Additionally, solving all access barriers does not ensure that RHD outcomes will be improved: Low quality of care potentially wastes scarce resources and undermines trust in health care systems. Patient uptake of penicillin prophylaxis and appropriate use of medications like warfarin have been documented to be low, even among those receiving care at tertiary academic hospitals.[26] The findings of this study are a necessary starting point for developing national RHD strategies and policies, but achieving universal health coverage in the area of RHD will require additional assessments and health system reforms and policies that are more comprehensive.

We conclude that the Ugandan health system's readiness to prevent and treat RHD is currently low. The country's national RHD strategy and policies, which are currently under development, should emphasize the following principles: (i) decentralization of rheumatic fever and RHD diagnostic capability to the district level, (ii) integration of curricular activities on RHD prevention, diagnosis, and treatment in schools of nursing, midwifery, and medicine, combined with in-service training for primary health workers, and (iii) strengthening of supply chains for benzathine penicillin G and other essential medicines to prepare for large increases in health service utilization that result from greater awareness and diagnostic capability. Efforts to scale up RHD interventions in Uganda provide opportunities to conduct prospective policy

and implementation research and identify best practices that can be transferred to other low-resource countries.

## Supporting information

**S1 Text. Supplementary methods and findings.** List of legends for S1 Text. **S1 Text Table A:** Organization of Uganda's health care system. **S1 Text Table B:** Indicators used to measure service readiness for three types of RHD interventions. **S1 Text Table C:** RHD service readiness by district and management authority. **S1 Text Table D:** Proportion of health facilities reporting availability of all RHD service readiness indicators. **S1 Text Fig A:** Districts included in the study. **S1 Text Fig B:** Availability of specific RHD primary prevention indicators, by district. Note: Green bars indicate average scores (out of total possible scores). **S1 Text Fig C:** Availability of specific RHD management indicators by district. Note: Green bars indicate average scores (out of total possible scores). **S1 Text Fig D:** Correlation of RHD service readiness score (overall) with general service readiness score obtained from the SARA instrument. Note: General health facility service readiness scores were calculated using instructions published in: WHO (2010). *Monitoring the building blocks of health systems*: *a handbook of indicators and their measurement strategies*. Geneva: World Health Organization. **S1 Text Fig E:** Barriers and enablers to RHD care identified during health work interviews. **S1 Text Fig F:** Summary matrix of key themes from health worker interviews.
(DOCX)

## Author Contributions

**Conceptualization:** David Watkins.

**Data curation:** Yoshito Kawakatsu.

**Formal analysis:** Emma Ndagire, Yoshito Kawakatsu, Hadija Nalubwama, Rachel Sarnacki.

**Funding acquisition:** Andrea Beaton, Craig Sable, David Watkins.

**Investigation:** Hadija Nalubwama, Jenifer Atala, Jafesi Pulle, Rakeli Kyarimpa, Rachel Mwima, Rosemary Kansiime.

**Methodology:** Yoshito Kawakatsu, Hadija Nalubwama, David Watkins.

**Project administration:** Emmy Okello, Peter Lwabi.

**Supervision:** Emmy Okello, Peter Lwabi, Andrea Beaton, Craig Sable, David Watkins.

**Writing – original draft:** Emma Ndagire.

**Writing – review & editing:** Emma Ndagire, Yoshito Kawakatsu, Hadija Nalubwama, Jenifer Atala, Rachel Sarnacki, Jafesi Pulle, Rakeli Kyarimpa, Rachel Mwima, Rosemary Kansiime, Emmy Okello, Peter Lwabi, Andrea Beaton, Craig Sable, David Watkins.

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
