## [Decision Letter · Decision Letter 0]

26 Oct 2020

Dear Dr. Watkins,

Thank you very much for submitting your manuscript "Examining the Ugandan Health System’s Readiness to Deliver Rheumatic Heart Disease-Related Services" for consideration at PLOS Neglected Tropical Diseases. As with all papers reviewed by the journal, your manuscript was reviewed by members of the editorial board and by several independent reviewers. In light of the reviews (below this email), we would like to invite the resubmission of a significantly-revised version that takes into account the reviewers' comments. 

Thank you for submitting this importance study to Plos NTD. I agree that methods and tools described here can become a model to be used by ministries of health in other countries to develop or expand their rheumatic heart disease programs. 

I would ask you to careful consider the suggestions of the reviewer, and also to discuss a point that is absent from your text. Availability and access to health care resources are very different things and most of your discussion refers to the availability of specific resources. 

To the best my knowledge, in Uganda, even where high complexity procedures are available (for example, surgery or valvoplasty), patients may do not have access to them because they would need to pay for the service. The WHA resolution is very clear that access is the major issue here, and I strongly suggest the authors to discuss a little more about this question. 

Universal health coverage seems to be essential to guarantee access to health care in all levels. There was an UN General Assembly high-level meeting on universal health coverage in September 2019 and the resolution calls on Member States to accelerate progress towards universal health coverage with a focus on poor, vulnerable and marginalized individuals and groups.

We cannot make any decision about publication until we have seen the revised manuscript and your response to the reviewers' comments. Your revised manuscript is also likely to be sent to reviewers for further evaluation.

Sincerely,

Guilherme L Werneck

Deputy Editor

Guilherme Werneck

Deputy Editor

Thank you for submitting this importance study to Plos NTD. I agree that methods and tools described here can become a model to be used by ministries of health in other countries to develop or expand their rheumatic heart disease programs. 

I would ask you to careful consider the suggestions of the reviewer, and also to discuss a point that is absent from your text. Availability and access to health care resources are very different things and most of your discussion refers to the availability of specific resources. 

To the best my knowledge, in Uganda, even where high complexity procedures are available (for example, surgery or valvoplasty), patients may do not have access to them because they would need to pay for the service. The WHA resolution is very clear that access is the major issue here, and I strongly suggest the authors to discuss a little more about this question. 

Universal health coverage seems to be essential to guarantee access to health care in all levels. There was an UN General Assembly high-level meeting on universal health coverage in September 2019 and the resolution calls on Member States to accelerate progress towards universal health coverage with a focus on poor, vulnerable and marginalized individuals and groups.

Reviewer's Responses to Questions

**Key Review Criteria Required for Acceptance?**

**Methods**

-Are the objectives of the study clearly articulated with a clear testable hypothesis stated?

-Is the study design appropriate to address the stated objectives?

-Is the population clearly described and appropriate for the hypothesis being tested?

-Is the sample size sufficient to ensure adequate power to address the hypothesis being tested?

-Were correct statistical analysis used to support conclusions?

-Are there concerns about ethical or regulatory requirements being met?

Reviewer #1: 1. The authors do not describe the methods for comparing the RHD service readiness score with the SARA-based general service readiness score.

2. The authors do not adequately detail how the quantitative data informed the qualitative data collection in this study they describe as explanatory mixed methods.

Reviewer #2: The objectives of the study were clearly articulated. This study did not require a testable hypothesis. The design of the study was appropriate to address the study objectives. The population of the study was clearly described and appropriate. There was no sample size estimation for the study. This study had both quantitate and qualitative methods of data collection. The sampling units were the health facilities, and convenience sampling was used to select these facilities, and the limitations discussed. 

The researchers used correct statistical analysis for the quantitative aspect of the study. 

The ethical and regulatory requirements were met.

**Results**

-Does the analysis presented match the analysis plan?

-Are the results clearly and completely presented?

-Are the figures (Tables, Images) of sufficient quality for clarity?

Reviewer #1: The results are clear and organized.

Reviewer #2: The results were clearly and completely presented, and the analysis presented matched the analysis plan. The figures are of sufficient quality.

**Conclusions**

-Are the conclusions supported by the data presented?

-Are the limitations of analysis clearly described?

-Do the authors discuss how these data can be helpful to advance our understanding of the topic under study?

-Is public health relevance addressed?

Reviewer #1: The conclusions and limitations match the data and methods.

Reviewer #2: The conclusions of the study are supported by the data. The authors discuss the limitations of the methods. They discuss how the data can advance the topic under study and the public health relevance.

**Editorial and Data Presentation Modifications?**

Reviewer #1: (No Response)

Reviewer #2: I recommend that this manuscript is accepted for publication.

**Summary and General Comments**

Reviewer #1: The authors present an important study documenting the readiness of a health system to provide care for prevention and management of RHD at primary and hospital level in Uganda. The authors undertook a major study involving surveying and site visits to about 400 health facilities and qualitative interviews among 36 healthcare providers. The authors developed an RHD service readiness survey tool and employed qualitative methods to better understand the health system gaps that need to be addressed to realize improved RHD care. The results are very relevant to the development of national RHD care policies and strategies in Uganda, and can be generalized to other low- and middle-income countries battling a high prevalence of RHD.

I have mainly minor/clarifying comments for the authors. Addressing these comments will improve clarity and flow of the manuscript.

Abstract

1. Methods. The authors report employing a “standardized instrument” for the health facility survey. Consider stating from where this instrument came or was adapted.

2. The sentence starting with “Health worker interviews perceived” should be rephrased or “perceived” should be replaced by another work such as “revealed” or “demonstrated.” It is the health workers who perceive, not the interviews themselves.

3. In the Conclusion section, is “education” the best word or is “specialized training” more representative of the message?

Author Summary

1. “how to address this problem” is slightly misleading – the study is about assessment of existing infrastructure/exposing health system gaps. While areas of need are identified, the study does not specifically posit or test strategies or frameworks for implementation.

2. The sentence beginning with “Our approach used” would benefit from rephrasing using less colloquial language.

3. “have everything they need” could be rephrased to read “fully equipped”.

Introduction

1. Page 1 lines 6-9. The authors enumerate 2 strategies to prevent RHD. However, primordial prevention is not mentioned. Primordial prevention of RHD through socioeconomic advancement is one of the primary drivers of the previously mentioned eradication of RHD in wealthy nations. Consider either mentioning primordial prevention, or emphasizing that the 2 strategies noted are those most amenable by the health system.

2. Page 1 line 10-11: can remove this sentence or move to later in paper to improve flow in introduction

Methods

1. Page 2-3 line 51-54. The authors mention districts were chosen because of specific features including representative of the general population and because existing studies were being performed. Does the active research programs influence health system/staff preparedness for RHD care?

2. Page 3, line 70. How many health facilities were invited from the census? What is the response rate?

3. Page 3, line 76. SARA acronym is the Service Availability and Readiness Assessment. The term “service” is omitted. Please correct.

4. Page 4, line 85. Please emphasize the data collection included both verbal/written response of respondents, as well as direct observations at the health facility. This is not clear as written.

5. Page 4, line 91. How did the study team ensure data entry was of high quality and free of entry errors?

6. Page 5 line 107-108: remove parentheses

7. Page 5, line 114. Was participation by providers voluntary? If so, this should be indicated. Currently there is a statement that facility representatives agreed to participate but not indicated that recruitment was free of coercion.

8. Page 5 line 115. Why did the authors decide to interview providers only at the highest level health facility in each of the 4 districts? Why not sample from a range of health facility types, including lower levels where most primary prevention activities would be conducted?

9. Page 5 line 119: “no connection” – does this refer to conflict of interest?

10. Page 5 line 123: Was English the primary language of interviewed health workers? It would be good to include a statement of understanding of health workers of interview questions and whether any translation to the local language was required.

11. Page 5 line 124. The authors describe this study as a sequent exploratory mixed methods study. Generally, the quantitative data will inform the qualitative data collection. In what way did the quantitative inform the development of the semi-structured interview guide? Or was this more of a triangulation design?

12. Page 6 line 134. The authors state they used a pre-written deductive codebook for the qualitative analysis. What prior conceptions or frameworks formed the basis for the pre-written codebook? The authors later reference the socioecological framework in the results.

13. Page 6. Ethics. Was there ethics approval at Children’s National Hospital – where the REDCap data were stored?

14. Do the authors have a definition of rural in terms of population? It may be helpful for the reader for this to be specified either in the text or in the caption of Table 1.

Results

1. Page 9 table 2. Point-of-care INR tests not available at hospitals. Are there other types of INR test platforms?

2. Page 10 line 197-198: please specify the name of the test (ASOT?)

3. Page 11 line 225. The authors report comparing RHD service readiness with general facility service readiness. This is not described in the methods. 

4. Page 12 line 236. Authors report “higher number of nurses in a health facility was also associated with a significantly higher RHD readiness score.” In table 3, the OR for this relationship is 1.0 with 95% CI 1.0, 1.0; p=.015. Is there a typo in the table, or is does the written statement not reflect the data.

Discussion

1. Page 15 line 300: “We documented low availability of RHD-related services in four representative districts…” As stated previously, I have reservations describing the 4 districts as “representative”. In fact, the authors state this later in the discussion (line 368-370).d

2. Page 16 line 316-317: public education is important but seems a bit tangential to the overall message as the study focuses on the health system. Suggest removing or moving to a later section in order to streamline the message in that particular section. The message about readiness is important and adding statements about public education detracts a bit from the overall message

3. Page 17 line 347: use of the word “pathology” is unclear here – does this refer to tissue pathology, laboratory services or treatment of RHD in general?

4. Page 17 line 350-362: Suggest abbreviating this paragraph to improve flow of the discussion

5. Page 17 line 359-361. Authors report that medical management of complicated RHD is not cost-effective in the absence of cardiac surgery. No reference is provided. Please provide a reference. I suspect more appropriate term instead of “cost-effective” would be “prudent”.

6. Page 18 line 371-374: parentheses can be removed here

7. One factor that may or may not be a source of bias in the survey and interviews was pressure by providers based at health centers/hospitals to make facilities seem more equipped, in other words better, than they actually were. Was this a limitation of the study?

8. Do the authors have a sense of whether the incomplete penicillin stock was due to frequent stock outs, refrigeration issues, or road/transportation issues? This would be interesting to add to the discussion.

9. Page 19 line 393. The authors earlier question the utility of pre-service vs. in-service training, but do not mention in-service training here. Continuing training is probably more important than pre-service training in maintaining high quality care.

Tables

1. Table 2: Does NA indicate that the variable did not exist at the particular health facility or that it was not expected to exist? This should be clarified. If the resource was expected to exist but not in stock, this should probably be represented by 0 rather than NA

Reviewer #2: This publication addresses a major public health problem in low income countries. The needs assessment done generated data that will guide the prevention of Rheumatic heart disease in Uganda. The study was well executed using WHO tools , and can be replicated in other countries. The manuscript was well written.

PLOS authors have the option to publish the peer review history of their article (what does this mean?). If published, this will include your full peer review and any attached files.

Reviewer #1: Yes: Gene F. Kwan, MD, MPH

Reviewer #2: No
---

## [Decision Letter · Decision Letter 1]

20 Jan 2021

Dear Dr. Watkins,

We are pleased to inform you that your manuscript 'Examining the Ugandan Health System’s Readiness to Deliver Rheumatic Heart Disease-Related Services' has been provisionally accepted for publication in PLOS Neglected Tropical Diseases.

Best regards,

Antonio Luiz Ribeiro

Guest Editor

Guilherme Werneck

Deputy Editor

Reviewer's Responses to Questions

**Key Review Criteria Required for Acceptance?**

**Methods**

-Are the objectives of the study clearly articulated with a clear testable hypothesis stated?

-Is the study design appropriate to address the stated objectives?

-Is the population clearly described and appropriate for the hypothesis being tested?

-Is the sample size sufficient to ensure adequate power to address the hypothesis being tested?

-Were correct statistical analysis used to support conclusions?

-Are there concerns about ethical or regulatory requirements being met?

Reviewer #1: (No Response)

**Results**

-Does the analysis presented match the analysis plan?

-Are the results clearly and completely presented?

-Are the figures (Tables, Images) of sufficient quality for clarity?

Reviewer #1: (No Response)

**Conclusions**

-Are the conclusions supported by the data presented?

-Are the limitations of analysis clearly described?

-Do the authors discuss how these data can be helpful to advance our understanding of the topic under study?

-Is public health relevance addressed?

Reviewer #1: (No Response)

**Editorial and Data Presentation Modifications?**

Reviewer #1: (No Response)

**Summary and General Comments**

Reviewer #1: The authors should be commended on this health facility census to evaluate readiness for RHD services in Uganda by both quantitative and qualitative means. I appreciate the authors' revisions and have no further suggestions.

PLOS authors have the option to publish the peer review history of their article (what does this mean?). If published, this will include your full peer review and any attached files.

Reviewer #1: **Yes: **Gene F. Kwan, MD, MPH

---

## [Editor Report · Acceptance letter]

5 Feb 2021

Dear Dr. Watkins,

We are delighted to inform you that your manuscript, "Examining the Ugandan Health System’s Readiness to Deliver Rheumatic Heart Disease-Related Services," has been formally accepted for publication in PLOS Neglected Tropical Diseases.

Best regards,

Shaden Kamhawi

co-Editor-in-Chief

Paul Brindley

co-Editor-in-Chief
